# PRE-TRAINING GOAL-BASED MODELS FOR SAMPLE-EFFICIENT REINFORCEMENT LEARNING

**Haoqi Yuan[1], Zhancun Mu[2], Feiyang Xie[2], Zongqing Lu[1,3†]**

[1]School of Computer Science, Peking University
[2]Yuanpei College, Peking University
[3]Beijing Academy of Artificial Intelligence

## ABSTRACT

Pre-training on task-agnostic large datasets is a promising approach for enhancing the sample efficiency of reinforcement learning (RL) in solving complex tasks. We present PTGM, a novel method that pre-trains goal-based models to augment RL by providing temporal abstractions and behavior regularization. PTGM involves pre-training a low-level, goal-conditioned policy and training a high-level policy to generate goals for subsequent RL tasks. To address the challenges posed by the high-dimensional goal space, while simultaneously maintaining the agent's capability to accomplish various skills, we propose clustering goals in the dataset to form a discrete high-level action space. Additionally, we introduce a pre-trained goal prior model to regularize the behavior of the high-level policy in RL, enhancing sample efficiency and learning stability. Experimental results in a robotic simulation environment and the challenging open-world environment of Minecraft demonstrate PTGM's superiority in sample efficiency and task performance compared to baselines. Moreover, PTGM exemplifies enhanced interpretability and generalization of the acquired low-level skills. Project page: https://sites.google.com/view/ptgm-iclr/.

## 1 INTRODUCTION

Deep reinforcement learning (RL) has achieved great success in solving sequential decision-making tasks (Silver et al., 2016; Vinyals et al., 2019; Hafner et al., 2020). However, many real-world domains such as indoor robotic tasks (Brohan et al., 2023; Myers et al., 2023) and open-world games (Team et al., 2021; Johnson et al., 2016) present significant challenges to RL. The complexity and long-horizon nature of these tasks make it difficult for RL to explore and receive positive rewards, thereby resulting in low sample efficiency. In recent years, we have increasing accessibility to vast datasets of robotic manipulations (Li et al., 2023) and human gameplay videos from the Internet (Baker et al., 2022; Fan et al., 2022). Pre-training on such datasets to improve RL has emerged as an important research topic.

These large-scale datasets are often not tailored for specific tasks. For example, VPT (Baker et al., 2022) gathers extensive data of human players playing the open-world game Minecraft, where the players explore freely rather than solve a specific task. There is potential to learn models of agent behaviors (Baker et al., 2022; Ramrakhya et al., 2023) and skills (Pertsch et al., 2021a;b) from these datasets to aid RL. We study pre-training low-level behaviors and skills, and then train high-level policies with RL for downstream tasks. This approach lies in hierarchical RL (Sutton et al., 1999), and provides temporal abstraction for the RL policies, thereby improving sample efficiency.

Existing methods (Pertsch et al., 2021a; Rao et al., 2021; Shah et al., 2021; Pertsch et al., 2021b; Shi et al., 2023) study pre-training low-level policies in low-dimensional RL environments (Fu et al., 2020) or narrow robotic domains (Gupta et al., 2020), but they have not scaled to high-dimensional, complex open-world environments (Li et al., 2023; Johnson et al., 2016) and large datasets (Baker et al., 2022; Fan et al., 2022). Some methods (Pertsch et al., 2021a;b; Shi et al., 2023) model low-

---

[†]Correspondence to Zongqing Lu <zongqing.lu@pku.edu.cn>.

level skills with latent variables using variational inference. However, they fail to model the complex action sequences, where the sequence length and the action space are large, in large datasets, e.g., Minecraft (Baker et al., 2022). Recent works (Baker et al., 2022; Lifshitz et al., 2023) reveal that policies with a transformer architecture (Vaswani et al., 2017) trained with behavior cloning on large datasets can effectively model various behaviors in Minecraft. Steve-1 (Lifshitz et al., 2023) trains a goal-conditioned policy that can exhibit various short-term behaviors conditioned on goals in Minecraft. Inspired by this, we introduce Pre-Training Goal-based Models (**PTGM**) for sample-efficient RL.

Our method, PTGM, pre-trains a goal-conditioned policy via behavior cloning and hindsight relabeling on a large task-agnostic dataset. To learn downstream tasks with RL, we train a high-level policy that outputs a goal at each step, with the goal-conditioned policy executing for several time steps in the environment based on this goal.

Training RL in a high-dimensional continuous action space is notably sample-inefficient (Lillicrap et al., 2015). If we follow the existing methods (Pertsch et al., 2021a;b) and attempt to mitigate this by reducing the dimensionality of the latent variables in the pre-trained model, the model will fail to learn diverse behaviors in the large dataset due to the decreased capacity. In our approach, the high-dimensional goal space presents a similar challenge. However, the generalization ability of the goal-conditioned policy pre-trained on large datasets enables us to compress the goal space without significantly diminishing the capacity. Consequently, we introduce a clustering approach to transform the goal space into a discrete high-level action space. Additionally, we propose to pre-train a goal prior model which predicts the distribution of future goals given the current state. The goal prior model provides an intrinsic reward with the KL divergence to the high-level policy in RL, regularizing the behavior of the agent to improve exploration.

We evaluate our method in a robotic manipulation environment Kitchen (Gupta et al., 2020) which requires solving subtasks sequentially, and the challenging open-world benchmark Minecraft (Fan et al., 2022) which contains diverse long-horizon tasks that are challenging for RL. PTGM outperforms baselines in terms of sample efficiency and success rates. Ablation studies demonstrate the necessity of each component in PTGM. Additionally, we demonstrate that PTGM has advantages in the interpretability and generalizability of the learned low-level skills.

In summary, the primary contributions of this work are:

- We propose pre-training goal-based models for RL, which holds advantages in the sample efficiency, learning stability, interpretability, and generalization of the low-level skills compared to existing methods.
- We propose the method of clustering in the goal space and pre-training the goal prior model, providing effective approaches for enhancing the sample efficiency of training high-level policies given a pre-trained goal-conditioned policy.
- Our experimental results validate the effectiveness of our method, demonstrating its capability to learn on diverse domains and solve the challenging Minecraft tasks efficiently.

## 2 RELATED WORK

**Pre-Training for RL.** This line of research can be categorized into two main settings: pre-training from task-specific datasets and pre-training from large task-agnostic datasets. For the former, imitation learning approaches (Gao et al., 2018; Ramrakhya et al., 2023) pre-train policies for initialization in RL, offline RL approaches (Lee et al., 2022; Zhu et al., 2023) pre-train policies and value functions, and transformers for RL (Wu et al., 2023; Sun et al., 2023; Escontrela et al., 2023; Xie et al., 2023) pre-train policies, transitions and state representations via sequence modeling. For the latter, Sermanet et al. (2018); Laskin et al. (2020); Aytar et al. (2018) pre-train state representations for image observations, Pertsch et al. (2021a;b); Shi et al. (2023); Rosete-Beas et al. (2023) learn low-level policies for temporal abstraction, and other works pre-train intrinsic rewards (Bruce et al., 2022; Zhou et al., 2023) or world models (Yuan et al., 2021; Seo et al., 2022). In this paper, we study pre-training low-level skills from task-agnostic datasets. Recent works (Baker et al., 2022; Lifshitz et al., 2023) demonstrate that imitation learning and goal-conditioned learning on a large task-agnostic dataset can acquire diverse skills effectively, motivating us to pre-train goal-based models from data.

**Goal-Conditioned RL.** Goal-conditioned RL (GCRL) (Liu et al., 2022) solves goal-augmented MDPs (Schaul et al., 2015) to achieve different goals. Schaul et al. (2015); Plappert et al. (2018); McCarthy & Redmond (2021) study training agents that can handle various goals and generalize across different goals in the multi-task learning setting. Andrychowicz et al. (2017); Chane-Sane et al. (2021); Zhu et al. (2021) address the challenges of RL with sparse rewards via GCRL. Nachum et al. (2018); Li et al. (2022) focus on temporal abstraction with goal-conditioned policies, developing hierarchical agents that can operate over high-level goals. Other works study goal representation learning in forms of images (Srinivas et al., 2018; Islam et al., 2022; Lifshitz et al., 2023) and languages (Myers et al., 2023; Cai et al., 2023) for GCRL or formulate offline RL with return-conditioned RL (Chen et al., 2021; Janner et al., 2021). Without RL, some methods perform goal-conditioned learning on datasets to build agents that can follow language instructions (Mezghani et al., 2023) or multi-modal prompts (Jiang et al., 2022). Our study falls within training goal-conditioned policies on datasets, providing temporal abstractions for downstream RL.

**Hierarchical RL.** Hierarchical RL (HRL) leverages temporal abstractions for sample-efficient learning in both single-task (Sutton et al., 1999; Kulkarni et al., 2016) and multi-task settings (Tessler et al., 2017; Veeriah et al., 2021), extensively integrating with model-based RL (Hafner et al., 2022), multi-agent RL (Mahajan et al., 2019), and imitation learning (Sharma et al., 2019b). We focus on methods that pre-train low-level policies for downstream RL. This includes unsupervised skill discovery with information-based objectives (Gregor et al., 2016; Sharma et al., 2019a; Strouse et al., 2021) and training skills from offline datasets (Pertsch et al., 2021a; Rao et al., 2021; Shah et al., 2021; Pertsch et al., 2021b; Shi et al., 2023). In this paper, we leverage a pre-trained goal-conditioned policy to enable temporal abstraction and study methods to improve sample efficiency for training the high-level policy with RL.

# 3 PRELIMINARIES

## 3.1 PROBLEM FORMULATION

A task can be formalized as a Markov Decision Process (MDP), defined by a tuple $M = (\mathcal{S}, \mathcal{A}, \mathcal{P}, \rho, R, \gamma)$ representing states, actions, the transition probability of the environment, the initial state distribution, the reward function, and the discount factor. Starting from the initial state, for each time step, the agent performs an action, then the environment transitions to the next state and returns a reward. Reinforcement learning (RL) learns a policy $\pi_\theta(a|s)$ to maximize the discounted cumulative reward $J(\theta) = \mathbb{E}_{\pi_\theta}[\sum_{t=0}^{\infty} \gamma^t R(s_t, a_t)]$. RL optimizes the policy by learning from online collected data $\{(s_t, a_t, r_t, s_{t+1})\}$. For partially observable MDPs (Kaelbling et al., 1998) with observations $o \in O$, we adopt the same notation as MDP, using $s_t$ to represent $o_{0:t}$.

We study tasks that are hard in exploration, where it is non-trivial for the agent to reach states that bring high rewards through random exploration. RL exhibits low *sample efficiency* on such tasks, meaning that it needs to collect a very large number of samples from the environment to improve task success rates. We assume access to a *task-agnostic* dataset $D = \{\tau = \{(s_i, a_i)\}_{i=0}^T\}$ collected in the same environment, in which the action sequences depict the non-trivial behaviors over time (i.e., $P_{\tau \sim D}(a_{t:t+k}|s_t) \neq \Pi_{i=t}^{t+k} P_{\tau \sim D}(a_i|s_t)$) generated by the agent (e.g. human players) while performing various tasks in the environment. Though trajectories in the dataset are sub-optimal for solving downstream tasks, the short-term behaviors $a_{t:t+k}$ in the dataset represent meaningful skills and can be stitched sequentially to accomplish a task (Badrinath et al., 2023).

We now formulate pre-training skills to provide temporal abstractions for RL as pre-training a model $P_\phi(a_{t:t+k}|s_t, z_t)$ using $D$, where diverse possible behaviors are modeled in the variable $z \in Z$. To train a task, RL can be performed on a high-level policy $\pi_\theta(z|s)$ which acts on a larger time scale $k$ and the pre-trained model $P_\phi$ decodes $a_{t:t+k}$ to act in the environment. The pre-trained model increases the probability of RL exploring towards task success by compressing the multi-step action space $A^k$ into a compact behavior space $Z$, thereby improving sampling efficiency.

## 3.2 GOAL-CONDITIONED POLICY

The pre-trained model is fixed during the RL phase, requiring it to have the capacity to model all the behaviors in the dataset and to decode action sequences accurately. We find that previous

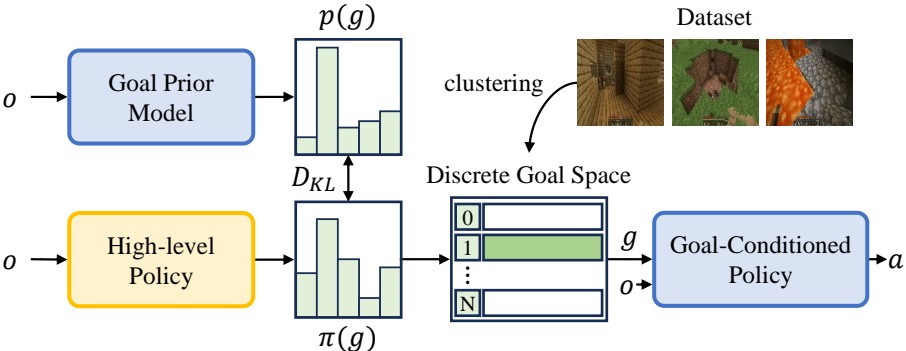

**Figure 1:** Overview of PTGM. A goal-conditioned policy and a goal prior model are pre-trained on the large task-agnostic dataset. The goals in the dataset are clustered into a discrete goal space. In downstream RL tasks, we train a high-level policy that outputs discrete actions to provide goals for the goal-conditioned policy, and regularize it with the KL divergence to the goal prior model.

methods (Pertsch et al., 2021a;b; Shi et al., 2023) that model the low-level behaviors with continuous latent variables struggle to model the complex distribution of action sequences in Minecraft datasets which feature large amounts of data, long action sequences to exhibit certain behaviors, and a large action space (more analysis in Appendix B.2). Recent work (Lifshitz et al., 2023) has demonstrated that training a goal-conditioned policy via behavior cloning with a transformer (Vaswani et al., 2017) architecture can stably learn various behaviors from these data. Therefore, we employ a similar approach to pre-train a goal-conditioned model.

A goal refers to the final state of a sub-trajectory in the dataset, representing the outcome of the short-term behavior. The goal-conditioned model $P_\phi$ uses the state space $S$ to be the space $Z$ and predicts the action for one step:

$$P(a_{t:t+k}|s_t, s^g) = \int_{s_{t+1},...,s_{t+k}} ds_{t+1} \ldots ds_{t+k} \prod_{i=t}^{t+k} P_\phi(a_i|s_i, s^g)\mathcal{P}(s_{i+1}|s_i, a_i), \qquad (1)$$

where $s^g \in S$ is the goal state. Starting from $s_t$, $P_\phi$ aims at reaching the goal state $s^g$ after executing the actions sampled from it for $k$ steps.

To learn $P_\phi$ from data, we use a variant of hindsight relabeling (Andrychowicz et al., 2017) to label each state-action sample in the dataset with a future state as the goal state. For a $k$-step subsequence $\tau = (s_t, a_t, \cdots, s_{t+k}, a_{t+k})$ in an episode, we label each sample $(s_i, a_i), t \leq i \leq t + k$ with the goal state $s^g = s_{t+k}$. We train $P_\phi$ with behavior cloning, minimizing the negative log-likelihood of action prediction:

$$\mathcal{L}(\phi) = \mathbb{E}_D \left[ - \log P_\phi(a_i|s_i, s^g) \right]. \qquad (2)$$

In practice, to enable the model to reach goals after different time steps, we randomly sample $k$ within a range. When states are compact vectors representing the physical state of objects in the environment, we use the raw state as the goal. When the environment is a POMDP with image observations, we use the embedding from a pre-trained image encoder as the goal. For instance, Steve-1 (Lifshitz et al., 2023) uses the embedding of $o_{t+k-16:t+k}$ from the MineCLIP's vision encoder (Fan et al., 2022) in Minecraft.

## 4 METHOD

In this section, we present the proposed method PTGM, utilizing the pre-trained goal-conditioned policy $P_\phi(a_t|s_t, s^g)$ to provide temporal abstractions for RL in downstream tasks. In RL, we train a high-level policy $\pi_\theta(s^g|s_t)$ which outputs a goal state to guide the low-level goal-conditioned policy $P_\phi$ to act in the environment for $k$ steps. To enhance the sample efficiency and stability of RL, we propose a goal clustering method and a pre-trained goal prior model. Figure 1 gives an overview of PTGM.

### 4.1 CLUSTERING IN THE GOAL SPACE

The goals $s^g \in S$ from the high-dimensional state space introduce a high-dimensional continuous action space for the high-level policy, making RL sample-inefficient. To tackle this challenge, we propose to cluster the states in the dataset to discretize the goal space, constructing a discrete action space for the high-level policy. We sample a large set of states from $D$, apply t-SNE (Maaten & Hinton, 2008) to reduce the dimension of states, and apply a clustering algorithm such as K-Means (Lloyd, 1982) to group similar goal states together and output $N$ clusters. The discretized goal space is represented with $G = \{i : s_i^g\}_{i=1}^N$, where $s_i^g$ is the goal state of the $i$-th cluster center. This converts the action space of the high-level policy into a discrete action space $A^h = [N]$ and constrains the high-level policy to output goals in the cluster centers.

We observe that compressing goal states into the discrete goal space does not significantly decrease the agent's model capacity to perform various behaviors. The reasons come from two aspects. Firstly, the clustering algorithm groups similar goals together. The cluster center can represent goals in its cluster that correspond to similar agent behaviors. Secondly, the goal-conditioned model pre-trained on the large dataset as the low-level policy can elicit generalizability on goals. The model can extract behavior information in the goal state, thereby generating correct behaviors even when the provided goal is distant from the current environment state. Given the same goal, the model can exhibit diverse behaviors in different states. These claims are substantiated in our experimental results in Section 5.4. Therefore, using the discrete goal space, the low-level policy still has the capacity to cover various behaviors in the dataset.

### 4.2 PRE-TRAINING THE GOAL PRIOR MODEL

In RL, the agent is able to perform smooth and reasonable behaviors within $k$ consecutive steps given a goal. However, the high-level policy lacks prior knowledge to provide reasonable goals, thereby should uniformly explore the goal space to learn the task. We propose to learn this prior knowledge from the dataset by pre-training a goal prior model, improving the sample efficiency and stability of training the high-level policy.

The goal prior model $\pi_\psi^p(a^h|s)$ has the same structure as the high-level policy, where $a^h \in A^h$ is the index of the goal cluster centers. This model is trained to predict the distribution of future goals given the current state, using the clustered goal space $G$. Similar to training the goal-conditioned model, we sample states and subsequent goal states $(s_t, s^g)$ from the dataset. In the discretized goal space $G$, we match the goal that is closest to $s^g$ based on cosine similarity $a^h = \arg\max_{i \in [N]} \left( \frac{s_i^g \cdot s^g}{\|s_i^g\| \cdot \|s^g\|} \right)$. The training objective for the goal prior model is to minimize the negative log-likelihood of goal prediction:

$$\mathcal{L}(\psi) = \mathbb{E}_D \left[ -\log \pi_\psi^p(a^h|s_t) \right]. \tag{3}$$

The pre-trained goal prior model acts as a regularizer for the high-level policy during RL, providing intrinsic rewards that guide the agent's exploration towards possible goals in the dataset.

### 4.3 REINFORCEMENT LEARNING WITH PTGM

Given the goal clusters $G$, the pre-trained low-level policy $P_\phi$, and the goal prior model $\pi_\psi^p$, we proceed with training the high-level policy using RL for downstream tasks. At each time step, the high-level policy $\pi_\theta(a^h|s)$ selects an index of the goal $s_{a^h}^g$ in the clustered goal space. The fixed low-level policy acts in the environment for $k$ steps conditioned on $s_{a^h}^g$. The high-level policy is updated based on the environment rewards and the intrinsic rewards from the goal prior model.

The overall objective for training the high-level policy is to maximize the expected return:

$$J(\theta) = \mathbb{E}_{\pi_\theta} \left[ \sum_{t=0}^{\infty} \gamma^t \left( \sum_{i=kt}^{(k+1)t} R(s_i, a_i) - \alpha D_{\mathrm{KL}} \left( \pi_\psi^p(a^h|s_{kt}) \| \pi_\theta(a^h|s_{kt}) \right) \right) \right], \tag{4}$$

where $t$ represents the number of steps for the high-level policy and $\alpha$ is a hyperparameter balancing the environmental rewards and the intrinsic rewards. By optimizing this objective, the high-level policy learns to select goals that lead to task success and align with the behaviors in the dataset, achieving sample-efficient RL with a discrete action space. In principle, any online RL algorithm can be used to train the high-level policy in downstream tasks.

## 5  EXPERIMENTS

### 5.1  ENVIRONMENTS AND DATASETS

We setup experiments on the following two challenging benchmarks with long-horizon tasks. More details about the environments and our implementations are presented in Appendix A.

**Kitchen.** A simulated robotic manipulation environment based on Gupta et al. (2020), where the agent controls a 7-DoF robot arm to manipulate objects in a kitchen. The dataset is provided in the D4RL benchmark (Fu et al., 2020), consisting of 150K transition samples. In pre-training, we use the environment state as the goal, which is a vector representing poses of the robot and objects. In downstream RL, we train on a long-horizon task consisting of 4 subtasks. The agent receives a sparse, binary reward for each successfully executed subtask. We use the SAC (Haarnoja et al., 2018) algorithm implemented in SPiRL (Pertsch et al., 2021a), replacing the max-entropy objective with minimizing KL-divergence to the goal prior model.

**Minecraft.** A popular open-world game that has been regarded as a challenging benchmark for RL (Guss et al., 2019; Baker et al., 2022; Fan et al., 2022). We adopt a video dataset of 39M frames labeled with actions introduced in Baker et al. (2022), which records the human players playing the game. In downstream RL, we use 5 tasks in the MineDojo simulator (Fan et al., 2022) which take thousands of steps to complete and are extremely difficult in exploration. The environment observations are images and the action space is keyboard and mouse operations discretized into $8641 \times 121$ choices. The agent receives a binary task success reward and the MineCLIP reward introduced in Fan et al. (2022). In pre-training, we use the MineCLIP embedding (Fan et al., 2022) of 16 consecutive frames as the goal. Then, we use PPO (Schulman et al., 2017) to train downstream tasks, optimizing a weighted sum of the extrinsic reward and the KL reward.

### 5.2  EVALUATION

To evaluate the performance of PTGM, we compare it to several baselines from recent works. More details on implementing these baselines are presented in Appendix B.

**SPiRL (Pertsch et al., 2021a).** This work pre-trains a sequential VAE (Zhu et al., 2020) for $k$-step action sequence $q(z|s_t, a_{t:t+k}), p(a_{t:t+k}|s_t, z)$ along with a skill prior $p_a(z|s_t)$, modeling skills with a continuous latent variable $z$. In RL, the high-level policy outputs a skill $z$, and then the action decoder $p$ decodes the action sequence and executes for $k$ steps. For Kitchen, we run the released code where $k = 10$. For Minecraft, there is a trade-off for selecting $k$. If we set $k = 100$ which is the same as PTGM, the model fails to reconstruct the long action sequences accurately. Otherwise, if we keep $k = 10$, the downstream RL can be less sample-efficient since less temporal abstractions are provided. Therefore, we run experiments on both settings, present the best result for each task in the paper, and leave the results of both settings in Appendix D.

**TACO (Rosete-Beas et al., 2023).** This work pre-trains a low-level policy $\pi(a_t|s_t, z)$ conditioned on the continuous latent variable $z$ from a learned skill posterior $q(z|\tau)$, regularizes $q$ with KL-divergence to a skill prior $p(z|s_t, s_T)$, and proposes offline RL methods to train the high-level policy. We take an online variant of TACO, where the skill prior $p$ is used to initialize and regularize the high-level policy for online RL. For each downstream task, we manually provide the task goal $s_T$.

**VPT-finetune.** VPT (Baker et al., 2022) is a foundation model for Minecraft that trains a behavior-cloning (BC) policy $\pi(a_t|o_{0:t})$ on a game playing dataset of 70K hours from the Internet. For Minecraft tasks, we train RL to either finetune the full model (Baker et al., 2022) or finetune the transformer adapters (Nottingham et al., 2023). We report the better results for each task in the paper and leave the results of both methods in Appendix D. Note that PPO from scratch fails on all

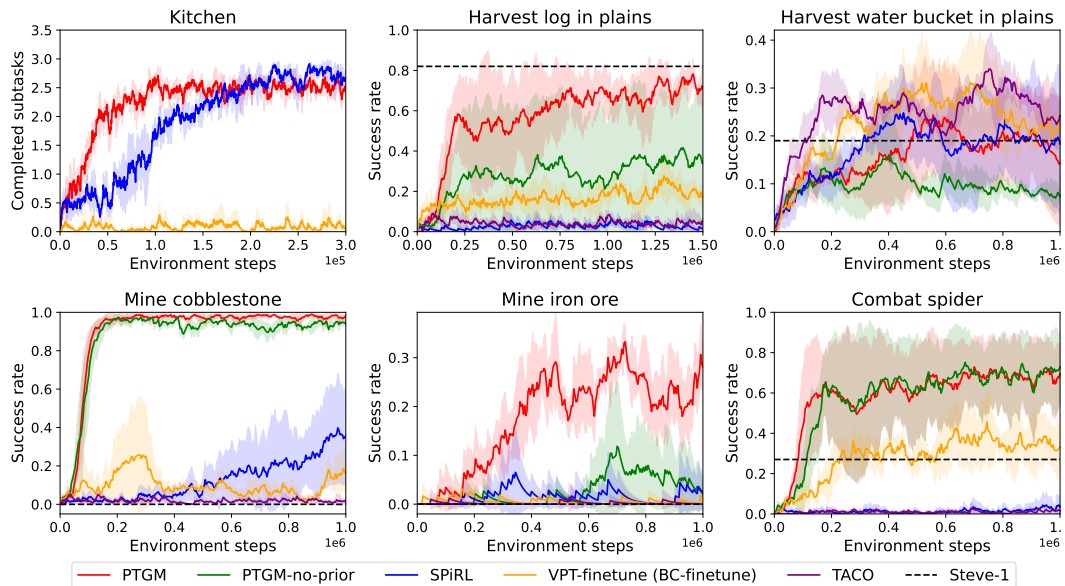

**Figure 2:** Training performance of PTGM against the baselines in all the tasks. The vertical axis indicates the number of completed subtasks in Kitchen and the success rate in the 5 Minecraft tasks.

downstream tasks in Minecraft according to Fan et al. (2022); Yuan et al. (2023). For Kitchen, we use the same approach to implement a baseline named **BC-finetune**.

**Steve-1 (Lifshitz et al., 2023).** This work builds an instruction-following agent in Minecraft. It first trains a goal-conditioned policy $\pi(a_t|o_{0:t}, g)$ on the contractor dataset, which is the same as the low-level policy in PTGM. Then, Steve-1 adopts a language-labeled dataset to map instructions to goals. We test the zero-shot performance of Steve-1 in our tasks by providing task instructions.

We measure the sample efficiency and task performance for all the methods with the training curves of success rates during RL. Figure 2 shows the performance of PTGM and all the baselines. In Kitchen, PTGM is able to solve 3 subtasks with high probability, exhibiting higher sample efficiency and comparable task success rates to SPiRL. Both PTGM and SPiRL outperform BC-finetune a lot, indicating that training RL with temporal abstraction provided by the pre-trained models improves sample efficiency significantly. In Minecraft, we observe that PTGM is the only method that achieves good success rates on all tasks after training for 1M environment steps, demonstrating its high sample efficiency and the strong capability to complete diverse surface and underground tasks. Results in most tasks show that the sample efficiency of PTGM greatly exceeds SPiRL, TACO, and VPT-finetune. PTGM is also the only method that completes the challenging Iron-ore task, which requires more than 1K exploration steps to obtain the rare item underground.

VPT-finetune fails in the Cobblestone and Iron-ore task. We believe one reason is that during RL, the policy without temporal abstraction may quickly forget the skill to repeat the same action consecutively to break a block. For Steve-1, we observe that it has the strong ability to cut trees; however, its performance on the other four tasks falls short. PTGM enhances the capabilities of Steve-1 by learning a high-level policy based on it.

We observe that SPiRL performs well in the simple domain Kitchen with small datasets, but struggles to learn challenging Minecraft tasks with large datasets and high-dimensional observations and actions, exhibiting worse performance compared to PTGM. We argue that the reasons for SPiRL underperforming PTGM in Minecraft include: the VAE in SPiRL struggles to reconstruct the long, high-dimensional action sequences; SPiRL trains high-level policies with RL in a continuous action space, while PTGM is capable of encoding rich and generalizable behaviors in a discrete action space, making downstream RL more sample-efficient. We verify the former reason in Appendix C and demonstrate in Section 5.4 that PTGM's discrete goals exhibit the rich behaviors and generalization capabilities mentioned in the latter reason. Due to the limited size and the lack of diverse behaviors at each state in the Kitchen dataset, it is not easy for PTGM to enhance the goal-conditioned policy with strong generalization capabilities, resulting in a smaller advantage in this task.

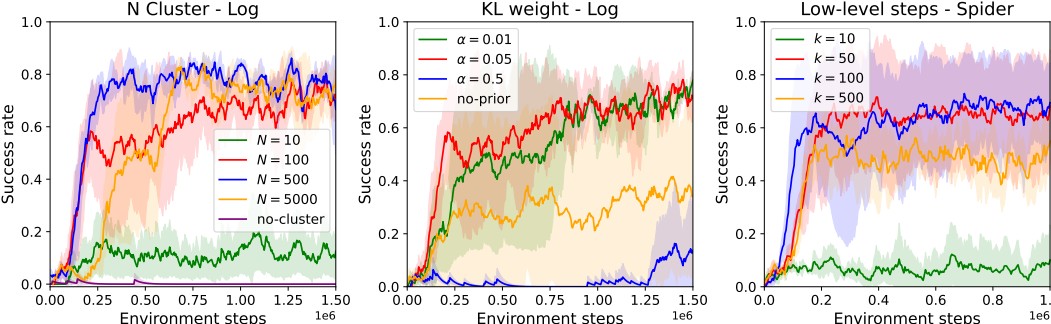

**Figure 3:** These figures show the curves of task success rates for the methods in the ablation study. The left figure shows PTGM with different numbers of goal clusters and without clustering (**PTGM-no-cluster**) in the Log task. The middle figure shows RL with different weights of the KL reward and RL without the goal prior model (**PTGM-no-prior**) in the Log task. The right figure shows RL with different numbers of low-level steps for each high-level action in the Spider task.

## 5.3 ABLATION STUDY

We conduct the ablation study on the three main components introduced in PTGM: clustering in the goal space, the KL reward provided with the goal prior model, and the temporal abstraction for RL. Figure 3 presents results in the Minecraft Log and Spider tasks. More results are presented in Appendix E.

For the number of clusters in the goal space, as shown in Figure 2, PTGM with 100 goal clusters is able to accomplish all the tasks. In Figure 3, with a goal space comprising 10 clusters, the agent fails to improve the task success rates. This is attributed to the non-existence of the tree-chopping behavior within the limited number of cluster centers. However, there remains a probability of task success, as the goal-conditioned policy can generalize and leverage goals associated with attacking other blocks in Minecraft to attack trees. We find that when the number of goal clusters is large, the performance of PTGM is robust to the change of this number. When the number increases to 500 and 5000, the high-level policy can still accomplish the task with high success rates. For $N = 5000$, the training curve rises a bit slower, indicating that the sample efficiency decreases due to the large high-level action space. But it still outperforms PTGM-no-cluster and SPiRL a lot. We find that PTGM-no-cluster, in which the high-level policy should output 512-dimensional continuous actions to match the original goal space, fails on the task.

For the intrinsic reward provided with the goal prior model, we find that a proper weight $\alpha$ for the KL reward improves both sample efficiency and task success rates for RL. RL with $\alpha = 0.01$ and $0.05$ outperform others and have low variance across different seeds. PTGM-no-prior exhibits a higher training variance than PTGM on the tasks of Log and Iron ore, as shown in Figure 2. We argue that without the KL reward, PTGM-no-prior suffers from inefficient random exploration in the large discrete goal space, resulting in much larger variance across different seeds. We conclude that the KL reward with the goal prior model contributes to both sample efficiency and learning stability for RL. As shown in Figure 3, for experiments with a large KL reward of $\alpha = 0.5$, we find that the task success rate increases slower, which may be attributed to the high-level policy easily converging to the goal prior model and ignoring the task success reward.

For the level of temporal abstraction, we observe that when the number of steps for the low-level policy is set to a small value ($k = 10$), PTGM has worse sample efficiency. It is because, under such settings, the number of steps required by the high-level policy to complete the task increases, making the exploration for task success more challenging. When the number of steps exceeds 50, PTGM has great sample efficiency and high success rates, accomplishing the task of combating a spider within 200K environment steps. This illustrates the effectiveness of the goal-conditioned policy pre-trained on the large dataset in successfully completing goals that involve hundreds of steps. In contrast, as detailed in Appendix D, SPiRL with the number of low-level steps set to $100$ fails on several tasks. It is worth noting that with a large number of $k = 500$, PTGM converges to a lower performance compared with $k = 100$. We argue that in this case, the high-level policy cannot

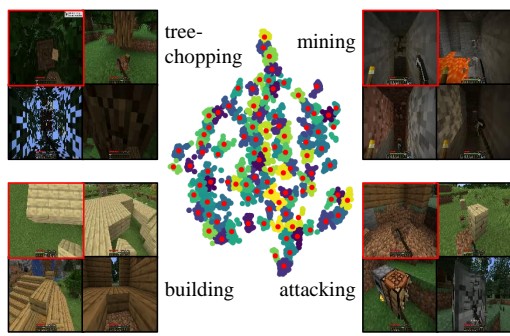

**Figure 4:** Visualization of goal clustering in Minecraft. Each group of 4 pictures is from the same cluster, where the picture with a red border is the cluster center. In the middle, each red dot means a cluster center.

| Test task | Sheep | Pig | Chicken |
|---|---|---|---|
| Success rate | 0.82 | 0.36 | 0.94 |
| Test task | Place | Water | Wool |
| Success rate | 0.65 | 0.16 | 0.44 |

**Table 1:** In each row, we pick a goal from the goal clusters and test the goal-conditioned policy in three tasks conditioned on this goal. For the first two rows, in the 16 frames corresponding to the goal, the agent is attacking a sheep. For the last two rows, in the 16 frames, the agent is building a house. In each test task, the target item (mobs or water) is initialized in front of the agent and we test for 100 episodes. The table shows the success rates.

switch many goals in an episode, making the task performance limited by the ability to solve the whole task conditioned on a single goal with the low-level controller. Figure 2 shows that $k = 100$ is a good choice for all the five Minecraft tasks, where the high-level controller is able to switch about 10 goals per episode.

## 5.4 INTERPRETABILITY AND SKILL GENERALIZATION

We believe that PTGM is sample-efficient not only because it provides temporal abstraction for RL, but also because it can encode rich behaviors in the compact discrete goal space. In Figure 4, we demonstrate that in the goal clusters, each cluster can represent an interpretable behavior of human players and samples in the same cluster exhibit similar behavior. The discrete goal space contains the behaviors of tree-chopping, mining, exploration, attacking, and building, which can make up the various skills required to play Minecraft.

Moreover, a single goal in the discrete goal space can be generalized to perform different skills, instead of representing a single skill only. As shown in Table 1, conditioned on the goal of attacking a sheep, the low-level policy can perform many similar behaviors including killing a sheep, killing a pig, and killing a chicken. Conditioned on the goal of house building, the low-level policy can place a block, collect water buckets, and harvest wool, since all these skills require right-clicking the mouse. Note that the test tasks have different terrains and backgrounds to the corresponding 16 frames of the goal. This demonstrates that, when provided with a single goal, the pre-trained goal-conditioned policy can generalize across various tasks and scenarios. This adaptability enriches the discrete goal space with diverse behaviors, thereby enabling the high-level policy to learn a variety of tasks within this compact goal space.

## 6 CONCLUSION

In this paper, we introduce PTGM to address the problem of sample-efficient RL in complex environments with large task-agnostic datasets. By employing a combination of the temporal abstraction provided with the pre-trained goal-conditioned policy, clustering in the goal space, and behavior regularization with the goal prior model, PTGM demonstrates superior performance across various tasks, notably in the challenging benchmark of Minecraft.

However, we recognize a few limitations and directions for future work. Firstly, the goal space in PTGM is inherently determined by the offline dataset, rendering the acquired skills susceptible to data bias. The agent may struggle to execute unseen skills, such as milking a cow in Minecraft, as the used dataset contains few instances of such behaviors. In the future, we plan to use larger Internet-scale datasets to enhance the capabilities of PTGM. Secondly, the efficacy of goal clustering relies on a good state representation that can compactly represent goals, especially in environments with image observations. We leave better goal representation learning for PTGM to future work.

ACKNOWLEDGMENTS

This work was supported by NSFC under grant 62250068.

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

# Appendices

## A ENVIRONMENTS AND IMPLEMENTATION DETAILS

### A.1 KITCHEN

The Kitchen environment features a kitchen scenario and several interactive objects. The subtasks in the downstream task involve opening the microwave, relocating the kettle, activating a bottom burner, and switching on the light. We utilize the 'mixed' version of the dataset provided by D4RL. This dataset comprises trajectories from human operations, with none of the sequences completing all four tasks. The state is a 60-dimensional vector, representing the positions and velocities of the 7-DoF robot arm and all the objects. The action is a 9-dimensional vector. A reward of $+1$ is given upon the successful completion of a subtask.

In pre-training, we set $k = 20$ for goal relabeling. We represent the goal with a 21-dimensional vector from the state, representing the positions of objects and the robot arm. In clustering, we apply t-NSE to reduce the goals to two dimensions, and then use KMeans++ to obtain 50 clusters. The goal-conditioned policy, the goal prior model, and the high-level policy are 5-layer MLPs with a hidden layer dimension of 128. The goal-conditioned policy and the goal prior model are trained until convergence and we select the model with the lowest validation loss.

In RL, we optimize the high-level policy, which is initialized with the weights of the pre-trained goal prior model. We use the SAC algorithm. To implement the KL reward, we replace the objective of entropy maximization in SAC with minimizing KL divergence to the goal prior model. Table 2 lists the hyperparameters of SAC.

**Table 2:** Hyperparameters of SAC.

| Name | Symbol | Value |
|---|---|---|
| Discount factor | $\gamma$ | 0.99 |
| Initial KL reward weight | $\alpha$ | 1 |
| KL reward target | $\delta$ | 0.7 |
| Low-level policy steps | $k$ | 20 |
| Rollout buffer size | -- | 1e6 |
| Training epochs per iteration | -- | 10 |
| Optimizer | -- | AdamW |
| Batch size | -- | 256 |
| Learning rate | -- | 3e-4 |

### A.2 MINECRAFT

We present the task setup in Minecraft with the MineDojo simulator in Table 3. The observations are $128 \times 128$ images in first-person view. The actions are formed by a dictionary representing keyboard states in binary values and mouse movements in $x, y$ directions, which are finally processed into a discrete action space of $8641 \times 121$. The reward for each task consists of two parts: a reward of $+1$ is given upon task success; and a reward provided by the vision language model MineCLIP, which computes the similarity between the task description and the last 16 frames of observations.

For the dataset, we use 39M frames from the contractor dataset introduced in VPT, consisting of 4 subsets where the human players are asked to play for different tasks such as house building and obtaining diamond pickaxe. We randomly sample $k$ between $[15, 200]$ for goal relabeling. We partition 10K frames from the dataset as the validation set. All models are pre-trained for more than 10 epochs until convergence, and we select the model with the lowest validation loss.

In clustering, we convert the action dictionary into a 25-dimensional one-hot vector and concatenate the 512-dimensional MineCLIP embedding of consecutive 16 frames with the sum of the corresponding 16 one-hot vectors of actions to form the goal state. Incorporating action information

**Table 3:** Settings for the five downstream tasks in Minecraft. *Language Description* for each task is used in both computing MineCLIP reward and testing the baseline of Steve-1. *Initial Tools* are provided in the inventory at each episode beginning. *Max Steps* is the maximal episode length.

| Task | Language Description | Initial Tools | Max Steps |
|---|---|---|---|
| Harvest log in plains | "Cut a tree." | -- | 2000 |
| Harvest water bucket in plains | "Find water, obtain water bucket." | bucket | 2000 |
| Mine cobblestone | "Obtain cobblestone." | wooden pickaxe | 500 |
| Mine iron ore | "Obtain iron ore." | stone pickaxe | 2000 |
| Combat spider | "Combat a spider in night plains." | diamond sword | 500 |

improves the distinction between different behaviors in Minecraft. We apply t-SNE to reduce the goals to two dimensions, and then use KMeans++ to obtain 100 clusters.

For the network architecture, the goal-conditioned policy, the goal prior model, and the high-level policy all use the architecture of VPT-2x model. It adopts an Impala CNN for image processing, which is comprised of three subsequent stacks with number of channels $\{16, 16, 32\}$ followed with a linear layer, converting the $128 \times 128 \times 3$ image into a 2048-dimensional vector. Then, the sequence of image embeddings are fed into a causal Transformer consisting of 4 subsequent causal residual transformer blocks, which can process sequence with a maximal length of 128. The Transformer output are 2048-dimensional vectors, providing state representations given past sequence of images. In the goal-conditioned policy, for each timestep, the MineCLIP embedding of the goal image is mapped to a 2048-dimensional vector with a linear layer and then added to the output of the image encoder to augment the input image with goal information. Given the Transformer outputs, a linear layer is followed to output $8641 \times 121$ categorical actions and another linear layer is used to output values. For the goal prior model and the high-level policy, we use a linear layer to output categorical actions with the number of goal clusters instead.

During pre-training, all the model parameters in the goal-conditioned policy and the goal prior model are optimized. In downstream RL, for the high-level policy, we use the pre-trained image encoder and transformer from VPT for feature extraction, and only optimize the MLPs in the transformer output layer, the action head, and the value head during RL. The action head and the value head are randomly initialized to provide a uniform prior in the goal space.

We use the PPO algorithm in RL. Table 4 lists the hyperparameters of PPO. We train each task for 3 random seeds.

**Table 4:** Hyperparameters of PPO.

| Name | Symbol | Value |
|---|---|---|
| Discount factor | $\gamma$ | 0.999 |
| KL reward weight | $\alpha$ | 0.05 |
| Low-level policy steps | $k$ | 100 |
| Rollout buffer size | -- | 40 |
| Training epochs per iteration | -- | 5 |
| Optimizer | -- | AdamW |
| Learning rate | -- | 1e-4 |
| GAE lambda | $\lambda$ | 0.95 |
| Clip range | -- | 0.2 |

## B    DETAILS FOR BASELINES

### B.1    VPT-FINETUNE

We adopt two methods to finetune VPT with RL, named VPT-adapter and VPT-full. VPT-adapter uses the VPT-3x model with 0.5B parameters. The overall architecture is mentioned in Section A.2 while the number of channels in Impala CNN are increased to $\{64, 128, 128\}$ and the hidden dimensions in Transformer are increased to 4096. We initialize the model with the pre-trained VPT-3x model and only optimizes the parameters in Transformer adapters and the value head during RL. VPT-full uses the VPT-2x model with 248M parameters, whose architecture is described in Section A.2. We initialize the model with the pre-trained VPT-2x model and optimizes all the parameters during RL. To alleviate the issue of catastrophic forgetting during RL, we add a KL loss between the output of the training policy and the initialized VPT model in PPO.

For the baseline of BC-Finetune in Kitchen, we use 5-layer MLPs with hidden layer dimensions of 128 to be the actor and the critic. The actor is pre-trained on the Kitchen dataset with behavior cloning until convergence. Then we finetune the actor-critic with SAC implemented in SPiRL, using the same configurations in Table 2.

### B.2    SPIRL

To implement SPiRL in Minecraft, we train SPiRL with different action sequence lengths, named SPiRL-10 and SPiRL-100. For SPiRL-10, we represent the state with the 128-dimensional MineCLIP embedding of the past 16 frames. The skill prior model is a 6-layer MLP with a hidden layer dimension of 128, which takes the current state embedding as input and outputs a 20-dimensional Gaussian distribution. In the autoencoder for action sequences, we use LSTMs to process sequence information. Concretely, the skill encoder takes the first state and the subsequent 10 actions as input. The action sequence is processed with an LSTM encoder with 128-dimensional hidden layers followed with a Linear layer to output a latent skill distribution with a dimension of 20. In the action decoder, an LSTM decoder with 128-dimensional hidden layers takes the latent skill and the state as input and decode the 10 actions. For SPiRL-100, we adopt the similar architecture but employ a larger model with a hidden layer dimension of 256 and a 40-dimensional latent skill space to increase the model capacity for generating longer sequences. To pre-train the model, we use a combination of the negative log-likelihood of action prediction, the KL divergence of the skill encoder to a unit Gaussian prior, and the negative log-likelihood of the skill prediction from the skill prior model. We set the weight of KL divergence to 10.0 to acquire the best result. In downstream RL, we use the high-level policy architecture described in Section A.2 and use a 20-dimensional Gaussian action head. We train PPO with a weight of 0.05 for KL reward.

In Kitchen, we adopt the official implementation of SPiRL.

We analysis the difficulty in extending SPiRL to the Minecraft domain which features **large-scale datasets** (with large amounts of data and diverse behaviors) **+ challenging open-world tasks** (with high-dimensional observation and action spaces, require long action sequences to execute certain behaviors). The reasons why SPiRL fails are threefold:

- SPiRL models behaviors in compact Gaussian latent variables $p(z|s_t, a_{t:t+k})$ regularized with KL divergence, which cannot accurately distinguish diverse behaviors in the large-scale Minecraft dataset.

- The low-level policy in SPiRL decodes multi-step actions at one state $\pi(a_{t:t+k}|s_t, z)$, which cannot accurately reconstruct the high-dimensional long action sequences in Minecraft. Thus, it fails to execute many complicated skills in Minecraft (e.g. breaking a log requires taking more than 10 attack actions repeatedly).

- In SPiRL, the high-level policy acts in the continuous skill space $z$, making downstream RL inefficient (especially for long-horizon tasks in Minecraft).

To address these issues respectively, we proposed PTGM that: 1. it models behaviors with goals (states in the dataset); 2. the goal-conditioned low-level policy $\pi(a_t|s_t, s^g)$ learns one-step action prediction on each state; 3. the high-level policy acts in the discretized goal space.

### B.3  TACO

We implement a variant of TACO that adopts offline skill pre-training and online RL. In Minecraft, we also use the VPT-2x backbone to implement the models. In the skill posterior, we use the last-token output in the Transformer to represent the whole trajectory and add a 2-layer MLP with a hidden layer size of 128 to extract a 20-dimensional Gaussian distribution. The low-level policy concatenates the state representation from the Transformer and the latent skill sampled from the skill posterior, using a 2-layer MLP with a hidden layer size of 128 to predict actions. The skill prior takes the state representation from the Transformer and the image representation of the last frame from the Impala CNN, using a 2-layer MLP with a hidden layer size of 128 to predict the latent skill distribution. During pre-training, we set the weight of KL loss to 1 to make both KL loss and action prediction loss converge to reasonable values.

For downstream RL, our high-level policy has the same architecture to the skill prior and we initialize it with the pre-trained skill prior. For each task, we manually pick an image in the dataset that demonstrates task success to provide a fixed task goal for both the skill prior and the high-level policy. We use PPO to train the high-level policy, setting the weight of KL reward to 0.01.

## C  RESULTS FOR PRE-TRAINING

Figure 5 shows the training loss of PTGM on the Minecraft dataset, with different numbers of goal clusters. Figure 6 shows the average reconstruction loss for each action in training SPiRL on the Minecraft dataset. The reconstruction loss of SPiRL-100 is significantly higher than that of SPiRL-10 after convergence, indicating that SPiRL has poor accuracy in modeling long action sequences of 100 steps in Minecraft.

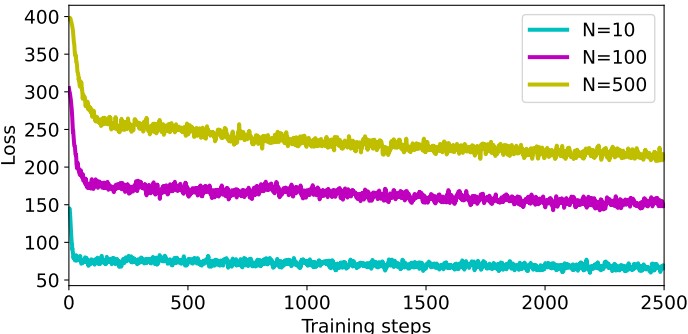

**Figure 5:** The training loss for pre-training the goal prior model in PTGM on the Minecraft dataset, with different numbers of goal clusters.

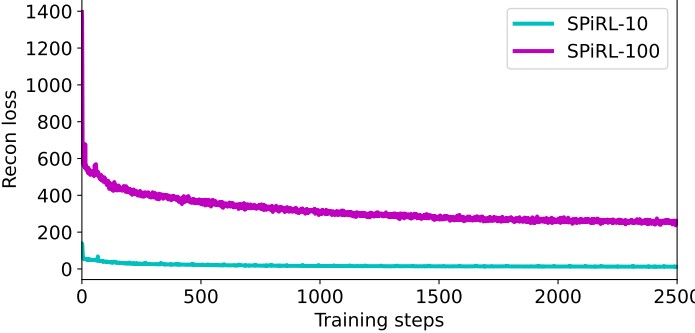

**Figure 6:** The action reconstruction loss for SPiRL-10 and SPiRL-100 on the Minecraft dataset.

# D   ADDITIONAL RESULTS FOR BASELINES

Figure 7 shows results for the baseline methods SPiRL-10, SPiRL-100, VPT-adapter, and VPT-full.

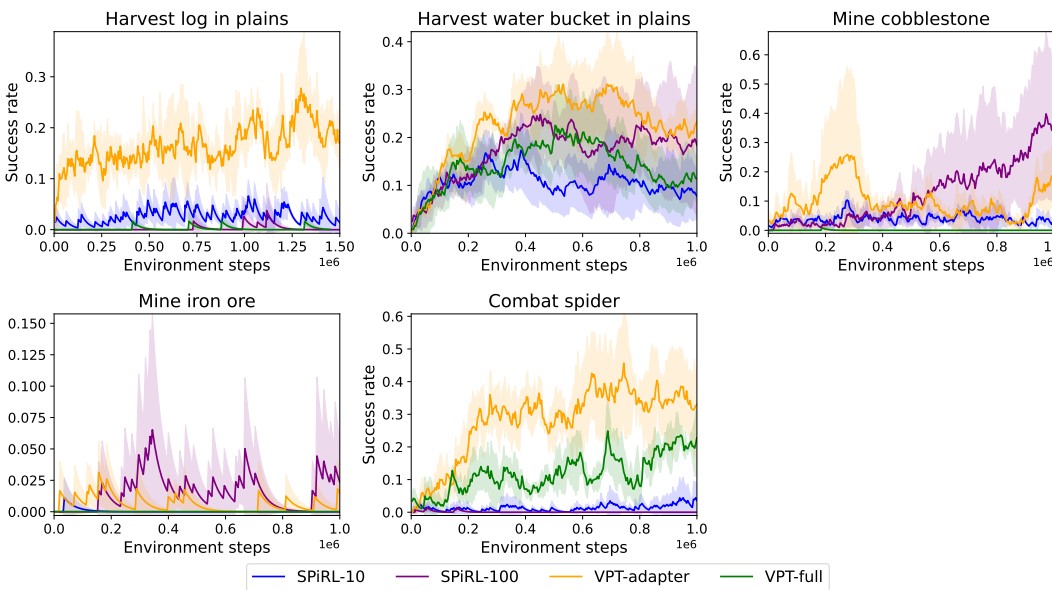

**Figure 7:** The RL training curves of SPiRL and VPT-finetune in Minecraft tasks. The vertical axis represents the success rate and the horizontal axis represents the number of steps in the environment. SPiRL-$k$ means the action decoder takes $k$ actions in the environment for each high-level action. VPT-adapter means finetuning the transformer adapters only and VPT-full means training the whole VPT model.

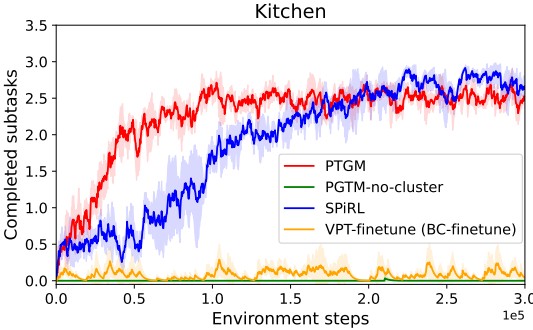

**Figure 8:** Results for different methods in Kitchen. PTGM-no-cluster means using the continuous goal space without clustering as the action space in the high-level policy.

Figure 8 shows results for different methods in Kitchen. BC-finetune pre-trains a behavior-cloning low-level policy on the Kitchen dataset and finetunes it with SAC. Its failure is related to the lack of temporal abstraction for RL and the forgetting issue during RL finetuning. (Pertsch et al., 2021a) also reports the similar results of this baseline in Kitchen.

Though the dimension of the goal in Kitchen is relatively lower (21 dimensions), we find that PTGM-no-cluster fails in this task. We observe that, with the continuous action space, the high-level policy should learn to output valid goals that lay in the manifold of states from the dataset to make the low-level policy perform reasonable behaviors. On the contrary, for PTGM with discrete goal clusters, the output of the high-level policy is always a valid goal (in the clusters), making RL efficient. Though the KL reward encourages PTGM-no-cluster to output goals close to the goal prior, it cannot make the output accurate enough due to the Gaussian sampling of the action head.

# E  ADDITIONAL RESULTS FOR THE ABLATION STUDY

Here, we present additional results of the ablation study in Minecraft tasks.

According to Figure 9, we draw some conclusions on the selection of cluster numbers:

- With $N = 10$, the agent fails to improve the task success rates, due to the low capacity of behaviors in the small goal space. But the task success rates are non-zero, due to the generalization ability of the low-level policy.

- With an increasing number of goal clusters, the performance of PTGM is robust, while the sample efficiency slightly decreases due to the larger high-level action space. In Spider, PTGM with $N = 5000$ underperforms $N = 100$ and $500$ in sample-efficiency and training stability.

- PTGM-no-cluster fails on all tasks, which is caused by ineffectiveness of RL acting in high-dimensional continuous action spaces.

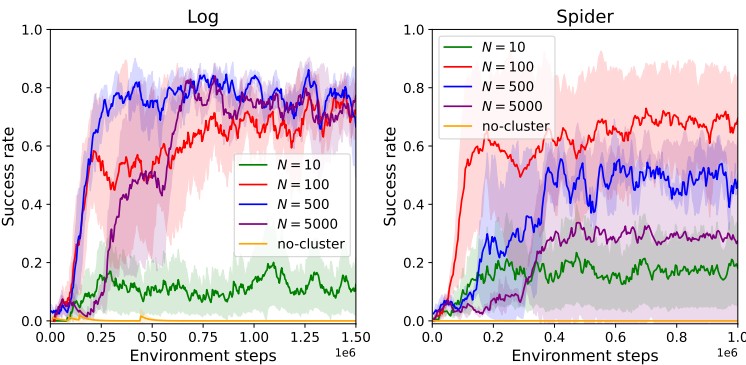

**Figure 9:** Results for the ablation study of PTGM with different numbers of goal clusters and without clustering (**PTGM-no-cluster**) in Minecraft tasks.

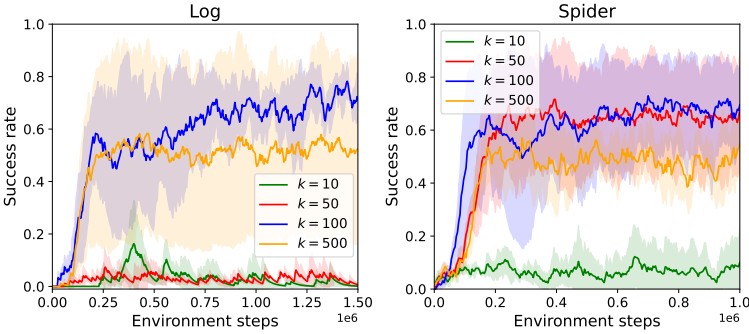

**Figure 10:** Results for the ablation study of PTGM with different numbers of low-level steps for each high-level action in Minecraft tasks.

According to Figure 10, we draw some conclusions on the selection of low-level step numbers:

- With a small number of low-level steps ($k = 10$), the behavior is offloaded more to the high-level policy, resulting in worse sample efficiency of RL.

- With an increasing number of low-level steps, PTGM shows better sample efficiency and success rates. $k = 100$ is a good choice for all the Minecraft tasks, where the high-level controller is able to switch about 10 goals per episode.

- With $k = 500$, PTGM converges to a lower performance compared with $k = 100$ because the behavior is offloaded too much to the low-level policy. In this case, the high-level policy cannot switch many goals in an episode, making the task performance limited by the capability of the low-level controller conditioned on a single goal.

Since prior work Pertsch et al. (2021a); Rosete-Beas et al. (2023) usually initialize the high-level policy with some pre-trained models for downstream learning, we also conduct ablation study that initialize the high-level policy with the goal prior model for downstream RL. According to Figure 11, PTGM-prior-init outperforms PTGM on harvesting logs, but has worse performance in other Minecraft tasks. We find that since chopping trees is the most frequent behavior in the Minecraft Contractor dataset Baker et al. (2022), the behavior-cloning models learned from such data can draw higher probabilities on harvesting logs than other behaviors. Thus, PTGM-prior-init is biased to the task of harvesting logs and fails to explore for other tasks in which the goal prior model draws low probabilities on the task-relevant goals. On the contrary, PTGM adopts a uniform goal prior for exploration, showing strong capabilities in solving out-of-distribution tasks.

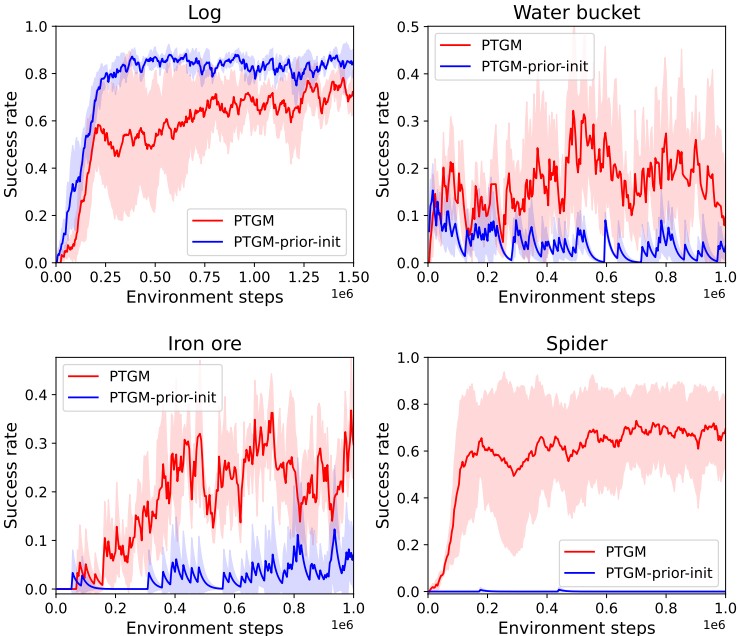

**Figure 11:** Ablation study for PTGM that initializes the high-level policy with the pre-trained goal prior for RL (**PTGM-prior-init**) in Minecraft tasks.

# F ADDITIONAL RESULTS FOR GOAL CLUSTER VISUALIZATION

Figure 12 provides additional results on goal clusters visualization, where we observe that samples in the same cluster exhibit similar behavior.

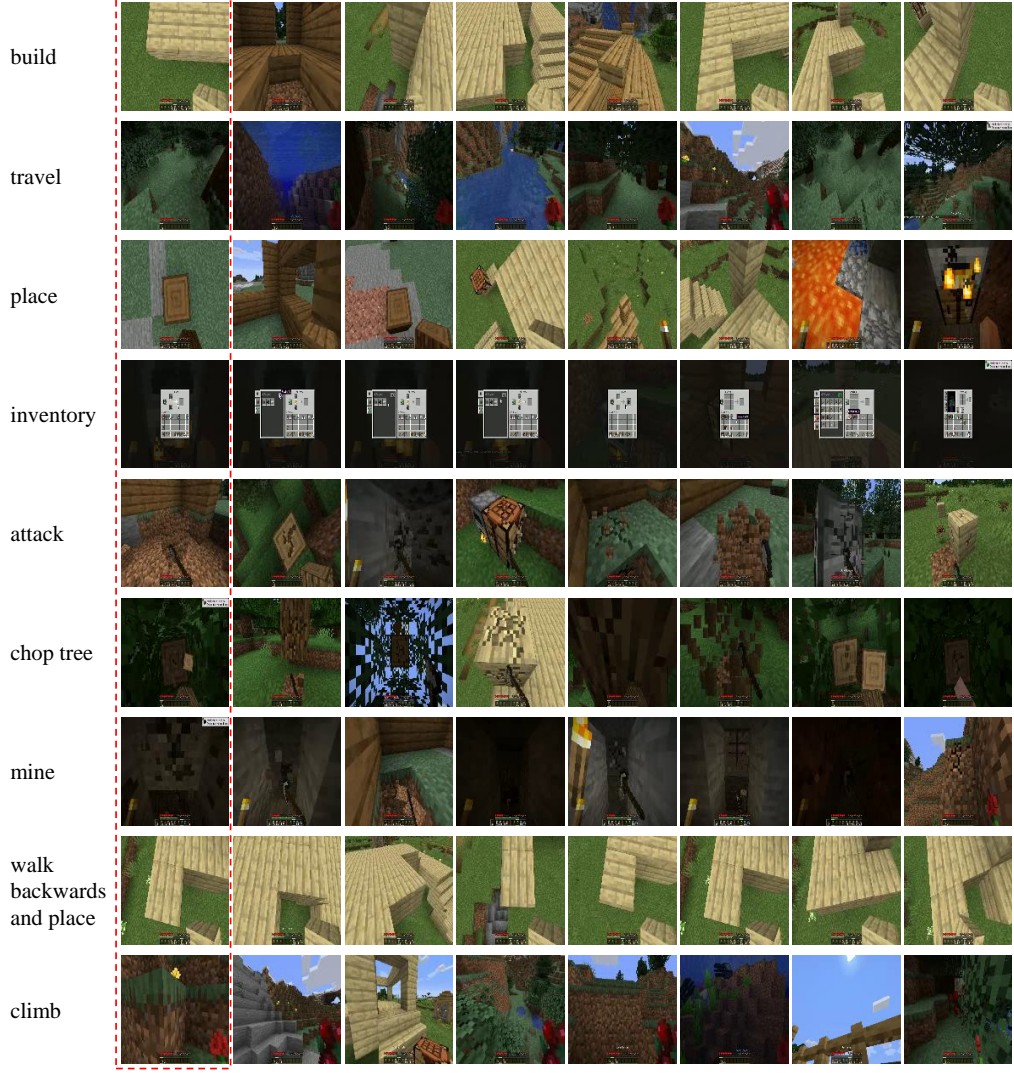

**Figure 12:** Additional results on goal clusters visualization in Minecraft. In each row, images are from the same cluster and the first image in the dotted red border is the cluster center. We manually annotate the semantic behavior represented by the images on the left.

