# OpenReview forum: "Pre-Training Goal-based Models for Sample-Efficient Reinforcement Learning"
_ICLR.cc/2024/Conference — ICLR 2024 oral_

### Official Review · Reviewer_WXZL · 2023-10-31

**Soundness:** 3 good
**Presentation:** 3 good
**Contribution:** 2 fair
**Rating:** 8
**Confidence:** 3

**Summary:**

This works presents Pre-Training Goal-based Models (PTGM), a hierarchical method to improve the sample efficiency of RL in complex environments for which large, task-agnostic datasets are available. PTGM first trains a low-level goal-conditioned policy via behavior cloning using the available dataset and then trains a high-level RL policy to generate goals for the low-level policy. Additionally, goal clustering is used to reduce the dimensionality of the goal space and enable the high-level policy to have a discrete action space while a goal prior model is used to guide the learning of the high-level policy. Experimental results in the robotic Kitchen environment and Minecraft demonstrate PTGM's superior sample efficiency and task performance.

**Strengths:**

1.	**Clear writing and presentation**: The paper is well-written and generally easy to follow. It provides the right intuition and effectively builds up motivation where needed. The related work covers a good number of papers.

2.	**Strong results in challenging domains**: PTGM has strong performance in comparison to existing baselines in complex settings such as Minecraft. This indicates that with sufficient engineering work, it can be applied to complex, large-scale environments. These results are very promising.

3.	**Interesting results with respect to interpretability**: The proposed goal clustering method provides interesting insights into the properties of the task-agnostic dataset.

**Weaknesses:**

1. **Relatively complex method with many steps**: The presented method requires the learning of three separate networks: a low-level goal-conditioned policy, a high-level goal-generating policy and a goal prior model. This results in a lot of hyperparameters such as the appropriate horizon (k) as well as the weight on the intrinsic reward for the high-level policy ($\alpha$). Additionally, the compression of the goal into a low-dimensional space is also driven by heuristics and requires the appropriate choice for the number of goals to cluster into. Applying this algorithm into a new domain will not be straightforward and would require significant engineering work. However, I acknowledge that if an end-user puts in the engineering effort, then this method can have very strong performance in a new domain.

2) **Novelty**: PTGM draws on several ideas from existing work by Pertsch et al.[1]. It would be helpful to get a better understanding of the similarities and differences between the two methods to judge the novelty of the work.

**Questions:**

1.	**Section 3.1** says *‘We study tasks that provide binary rewards, offering a reward of +1 only upon reaching a non-trivial success state’*. However, the Minecraft results have a MineCLIP reward that appears to be dense. Since the majority of the results are in Minecraft, I would revise the above sentence.

2.	The low-level policy is learned purely from the dataset. However, the trajectories in the dataset could be generated by sub-optimal agents which could result in a sub-optimal low-level goal-conditioned policy. The assumptions of the dataset and the properties of the low-level policy obtained from that should be clarified further. On the same note, I am curious to know if fine-tuning the low-level policy with RL was explored.

3.	The caption in **Table 1** should be rephrased for clarity. In particular, *‘two rows’* is confusing as the actual data is contained in a single row.

4.	What are the different tasks used for the dataset collection in Minecraft? Are they the same as the downstream tasks that PTGM is ultimately evaluated on? In other words, is PTGM able to generalize to new tasks that are different from individual trajectories in the dataset?

5.	It would be interesting to see the number of different (unique) goals output by the high-level policy during a trajectory. For example, for a trajectory of length 1000 with k=100, are all 10 goals output by the high-level policy different?

6.	With reference to **2)** in weakness, what are the similarities and differences between PTGM and SPiRL [1]?

7.	Task and goal seem to be used interchangeably in the paper. For more clarity, it would be useful to clearly define the difference between the two terms in the *Problem Formulation* section.


[1] Pertsch, Karl, Youngwoon Lee, and Joseph Lim. "Accelerating reinforcement learning with learned skill priors." Conference on robot learning. PMLR, 2021.

---

> ### Author Response · Authors · 2023-11-20
> **Thanks for your review! Here, we respond to your comments and address the issues. We hope to hear back from you if you have further questions!**
>
> **Q1.** About the novelty of PTGM and the differences to SPiRL [1].
>
> **A1.** While both SPiRL and our work study pre-training skills on task-agnostic datasets for RL, we find issues in extending SPiRL-like methods to challenging domains and propose quite different technical approaches.
>
> We observe that SPiRL fails in domains that feature **large-scale datasets** (with large amounts of data and diverse behaviors) **+ challenging open-world environments** (with high-dimensional observation and action spaces, require long action sequences to execute certain behaviors). The reasons why SPiRL fails are threefold:
> - SPiRL models behaviors in compact Gaussian latent variables $p(z|s_t,a_{t:t+k})$ regularized with KL divergence, which cannot accurately distinguish diverse behaviors in the large-scale dataset.
> - The low-level policy in SPiRL decodes multi-step actions at one state $\pi(a_{t:t+k}|s_t,z)$, which cannot accurately reconstruct the high-dimensional long action sequences in open-world domains. Thus, it fails to execute many complicated skills in Minecraft (e.g. breaking a log requires taking more than 10 attack actions repeatedly).
> - In SPiRL, the high-level policy acts in the continuous skill space $z$, making downstream RL inefficient (especially for long-horizon tasks in high-dimensional environments).
>
> To address these issues respectively, PTGM provides novel approaches that:
> - It models behaviors with goals (states in the dataset).
> - The goal-conditioned policy $\pi(a_t|s_t,s^g)$ learns one-step action prediction on each state. Such a low-level policy can perform diverse skills accurately as presented in [2].
> - The high-level policy acts in the discretized goal space.
> To this end, the methods of goal space clustering and goal prior regularization are also our original contributions to implementing such a framework.
>
> We agree that this discussion is helpful for readers to understand our work better and add it to Appendix B.2.
>
> [1] Accelerating Reinforcement Learning with Learned Skill Priors, 2021
>
> [2] STEVE-1: A Generative Model for Text-to-Behavior in Minecraft, 2023
>
> **Q2.** Section 3.1 says ‘We study tasks that provide binary rewards, offering a reward of +1 only upon reaching a non-trivial success state’.
>
> **A2.** Thanks for pointing out our writing mistakes! We revised the above sentence since we use a MineCLIP reward in Minecraft.
> We claim in Section 5.2 (the **VPT-finetune.** paragraph) that such reward cannot lead vanilla RL to task success. MineCLIP only provides a high-reward when the target items mentioned in the task description are close in front of the agent [3]. The exploration in RL is difficult when the target is unseen.
>
> [3] CLIP4MC: An RL-Friendly Vision-Language Model for Minecraft, 2023
>
>
> **Q3.** The trajectories in the dataset could be generated by sub-optimal agents. Is fine-tuning the low-level policy with RL explored?
>
> **A3.** We assume that the task-agnostic dataset contains non-trivial behaviors generated by the agent (e.g. human players) while performing various tasks in the environment. Though trajectories in the dataset are sub-optimal for solving downstream tasks, the short-term behaviors $a_{t:t+k}$ in the dataset represent meaningful skills and can be combined sequentially to accomplish downstream tasks better. Recent work [4] also demonstrates that stitching short-term goals from sub-optimal data can lead to better performance. We revised the second paragraph in Section 3.1 to clarify it.
>
> Finetuning to further improve the performance of the low-level policy is promising and remains unexplored in the literature of pre-training for RL. In future work, we could pre-train the goal-conditioned policy with offline RL methods and finetune it with online data collected in downstream tasks.
>
> [4] Waypoint Transformer: Reinforcement Learning via Supervised Learning with Intermediate Targets, 2023
>
> **Q4.** The caption in Table 1 should be rephrased for clarity.
>
> **A4.** Thanks! We have revised this caption.
>
>
> **Q5.** What are the different tasks used for the dataset collection in Minecraft?
>
> **A5.** We actually use only one task-agnostic human gameplay dataset introduced in VPT [5] to pre-train models for all downstream tasks. This satisfies our problem formulation that low-level behaviors are learned from a large task-agnostic dataset and the high-level policy can reuse the pre-trained skills to learn arbitrary tasks.
> For downstream tasks, the behaviors of combatting spiders and harvesting water buckets rarely appears in the dataset. Our method can stitch and generalize the goals to accomplish these tasks.
>
> [5] Video PreTraining (VPT): Learning to Act by Watching Unlabeled Online Videos, 2022

---

> > ### Author Response · Authors · 2023-11-20
> > **Response to the remaining two questions.**
> >
> > **Q6.** About the number of unique goals output by the high-level policy during a trajectory.
> >
> > **A6.** We pick the last training checkpoints to test for 1. the number of unique goals picked per episode; 2. the probability that the high-level policy picks a goal that is different from the last selected goal. We test each checkpoint for 100 episodes and report average results in the Table below.
> >
> >  Task | Kitchen | Log | Water | Cobblestone | Iron | Spider
> >  ------------- | ------------- | ------------- | ------------- | ------------- | -------------  | -------------
> >  Unique goals  |  13.71  |  1.42  |  2.30  |  1.00  |  1.12  |  1.59
> >  Change probability | 0.57 |  0.11  |  0.32  |  0.00  |  0.01  |  0.34
> >
> > In Kitchen, Log, Water, and Spider tasks, the learned high-level policy in PTGM picks various goals in an episode and is possible to switch to a different goal at each step. In Kitchen, since the downstream task is explicitly defined with four sub-tasks, the high-level policy should pick a large number of unique goals on average.
> > In the Cobblestone task, the high-level policy can simply choose to dig down without switching to other goals, thus the policy converges to always picking a unique goal. The Iron task is similar to Cobblestone but is much more challenging in its long-horizon nature. Though it does not require switching for diverse goals, only PTGM finds a good policy that chooses to dig down for many high-level steps and achieves a high success rate on this task (Figure 2).
> >
> > **Q7.** It would be useful to clearly define the difference between task and goal in the Problem Formulation section.
> >
> > **A7.** Thanks for your suggestion. In our paper, (downstream) tasks refer to long-horizon tasks defined with MDP. Goals refer to the last states of short-term behaviors in the dataset, which can represent low-level skills.
> > In problem formulation, we distinguish tasks and low-level behaviors and have not mentioned goals. We revised Section 3.2 to make the definition of goals clear.

---

> > > ### Comment · Reviewer_WXZL · 2023-11-21
> > > **Thank you**
> > >
> > > Thank you for taking the time to answer my questions, preparing additional results and modifying the paper to increase clarity. The number of unique goals sampled by the high-level policy in Minecraft is pretty low and downstream tasks in Minecraft that require more skills/stitching of more goals is a promising future direction to explore.
> > >
> > > Overall, I am happy with the work and have raised my score to reflect the same.

---

> > > > ### Author Response · Authors · 2023-11-22
> > > > **Thanks!**
> > > >
> > > > Thank you very much for raising your score! We appreciate your constructive review.
> > > > We agree with your comments about tasks with more goals and will explore this in future work.

---

### Official Review · Reviewer_VFeh · 2023-10-31

**Soundness:** 2 fair
**Presentation:** 2 fair
**Contribution:** 2 fair
**Rating:** 6
**Confidence:** 4

**Summary:**

This paper studies the pre-training goal-conditioned model to improve the sample efficiency in downstream RL training. The authors try to improve the learning of a goal generation model and propose to discretize the goal space into a fixed number of groups so that the goal generation model can handle high dimensional space tasks.

**Strengths:**

- Goal-conditioned RL is important for offline pre-training.
- The proposed method is easy to understand.

**Weaknesses:**

- The novelty of discretizing goal spaces is limited.
- The baselines are not sufficient. There are many goal-conditioned RL methods that can be applied to offline pre-training, including continuous goals [1] and discretized goals [2].  The included baselines are not well selected. *SPiRL* was proposed in 2021, and *VPT* is not goal conditioned. *Steve-1* is language-labeled which is used in quite different settings.
- I don’t think the visualization in Figure 4 can support that the clustered goals are meaningful. You can always find images from different clusters representing different behaviors. What I would expect is that images clustered into the same group should present similar patterns.

[1] Rosete-Beas, Erick, et al. "Latent plans for task-agnostic offline reinforcement learning." *Conference on Robot Learning*. PMLR, 2023.

[2] Islam, Riashat, et al. "Discrete factorial representations as an abstraction for goal conditioned reinforcement learning." *arXiv preprint arXiv:2211.00247* (2022).

**Questions:**

- Why does the clustering can produce meaningful goals? The clustering is done without any prior knowledge or inductive bias. If the image background is noisy, can the proposed clustering method work?

---

> ### Author Response · Authors · 2023-11-20
> **Thanks for your review! Here, we respond to your comments and address the issues. We hope to hear back from you if you have further questions!**
>
> **Q1.** The baselines are not sufficient. About using SPiRL, VPT, and Steve-1 as baselines.
>
> **A1.** Thanks for pointing out these related works! We have to emphasize that the main literature we study is "pre-training from data to accelerate RL", in which we find specific issues in learning skills from large task-agnostic datasets in open worlds (the third paragraph in Section 1) and propose the first goal-based approach PTGM to address this problem. While GCRL is related to our study, most works in GCRL are orthogonal to our work in the problem formulation (though we admit that techniques proposed in GCRL can be alternative implementations of some components in PTGM). TACO-RL[1] studies hierarchical approaches in fully offline RL; DGRL[2] studies goal representation learning for GCRL, which is not an issue considered in our study and can be used as an alternative method to provide goal representations for PTGM as discussed in Section 6. We added these related works in Section 2.
>
> Nevertheless, we take your comments and implement an online RL variant for TACO-RL to fit our settings. We update the results in Figure 2. While TACO holds competitive performance in harvest water bucket, it fails on other Minecraft tasks.
> We discuss that TACO has the following drawbacks compared with PTGM:
> - TACO provides a continuous high-level action space for downstream RL, which makes the high-level policy likely to output unseen latent skills and is harmful for sample-efficiency.
> - TACO requires manually picking a state representing the task goal for each downstream task. While the goal state for each task is deterministic in simple domains experimented in TACO-RL and SPiRL, the goal image for a Minecraft task can be diverse (e.g., for Harvest Log, there are various kinds of trees in Minecraft and the agent has diverse view angles), making it difficult to learn a good skill prior to enhance high-level RL.
>
> About the necessity of our selected baselines:
> - Though SPiRL was proposed in 2021, it is a strong baseline in the field of "pre-training skills from task-agnostic data for RL". Recent papers [3,4] under this setting also adopt this baseline.
> - VPT+RL is actually the strongest baseline in Minecraft which adopts the same setting of pre-training + RL finetuning. As reported in [5], it can solve the most challenging tasks in Minecraft with billions of environment steps.
> - Steve-1 uses the same goal-conditioned controller as our method and can serve as a strong baseline for Minecraft tasks. While Steve-1 maps goals to language instructions to build a multi-task agent, our high-level policy learned with RL can overcome its limitations that 1. cannot switch goals to perform hierarchical long-horizon control; 2. fails on tasks with unseen instructions.
>
> [1] Latent plans for task-agnostic offline reinforcement learning, 2022
>
> [2] Discrete factorial representations as an abstraction for goal conditioned reinforcement learning, 2022
>
> [3] Skill-Critic: Refining Learned Skills for Reinforcement Learning, 2023
>
> [4] Subwords as Skills: Tokenization for Sparse-Reward Reinforcement Learning, 2023
>
> [5] Video PreTraining (VPT): Learning to Act by Watching Unlabeled Online Videos, 2022
>
>
> **Q2.** About cluster visualization in Figure 4.
>
> **A2.** Thanks! We have modified Figure 4 to visualize images within the same cluster. The updated figure shows that images within the same cluster demonstrate goals of similar behaviors in Minecraft like tree-chopping or mining. We put more visualization results in Appendix F.
>
>
> **Q3.** Why does the clustering can produce meaningful goals without inductive bias?
>
> **A3**: As presented in Section 5.1, in Kitchen, we cluster on the environment states which contain physical positions of objects in the scene and are good representations to distinguish different goals; In Minecraft, we adopt the MineCLIP image embedding pre-trained from Internet-scale YouTube videos with captions, which may be able to capture the information of semantic items in Minecraft.
> While our study focuses on the skill pre-training + RL framework, learning good representations for goals (especially for image-based environments) can be complementary to our work, as we discussed in Section 6.

---

> > ### Comment · Reviewer_VFeh · 2023-11-22
> > **Thanks for the interpretation**
> >
> > Thanks for your effort in making this paper more understandable. I now value your motivation more and feel surprised that goal-based methods are not used in pre-training yet. However, the performance in this paper is not significant, and the clustering lacks a principled explanation of why it works. I would raise my score to marginal acceptance as the other reviewers like this paper more (I am probably too critical) but retain the concerns above.

---

> > > ### Author Response · Authors · 2023-11-22
> > > **Thanks!**
> > >
> > > Thank you very much for raising your score! We appreciate your constructive review.
> > > We will keep in mind your remaining concern and improve the clustering with better goal representation learning in future work.

---

### Official Review · Reviewer_NwB4 · 2023-10-31

**Soundness:** 3 good
**Presentation:** 3 good
**Contribution:** 3 good
**Rating:** 8
**Confidence:** 4

**Summary:**

This paper studies the problem of learning diverse and temporally extended behaviors using hierarchical RL with access to large, pre-existing datasets. The proposed method, PTGM, pre-trains (1) a goal-conditioned low-level policy using behavior cloning, and (2) a goal prior in a discrete space of goal clusters, both of which are extracted from a large pre-existing behavior dataset. After pre-training these two components, PTGM then trains a high-level RL policy to select goals for the goal-conditioned low-level policy to reach. A key technical contribution that sets this method apart from prior work is the use of a discrete goal space (by means of clustering), which greatly reduces the action space of the high-level policy that is learned via RL (as opposed to continuous goal embeddings). The authors argue that this discretization plays a key role in improving exploration during RL training.

Experiments are conducted on two task domains: a kitchen environment in which a robotic manipulator is tasked with manipulating multiple objects sequentially based on state inputs, and 5 visual tasks from MineDojo which is based on the open-world video game Minecraft. The authors find that PTGM generally improves over a number of recently proposed, seemingly strong baselines in both task domains.

**Strengths:**

- The problem setting is both interesting and timely, and will likely be of interest to the ICLR community. The paper is well written, positions itself wrt prior work, and is generally easy to follow, with a few exceptions (see *weaknesses* below).
- The technical contributions and design choices are generally well motivated and intuitive given the problem setting. I appreciate the relatively simple and data-driven approach to hierarchical policy learning which addresses two key challenges for this class of algorithms -- training stability and diversity of behaviors -- by leveraging existing datasets + only using RL when necessary.
- Experiments are conducted on fairly difficult tasks that span both state and image observations, improvements over baselines appear significant.

**Weaknesses:**

- Ablations are fairly limited and leave me with several unanswered questions that seem important to address given the technical contributions of the paper. Firstly, it is evident that too few clusters (10) fails to capture the diversity of the goal space, and that no clustering fails to learn at all. However, it is not clear to me what the breaking point would be in terms of number of clusters: would e.g. 5,000 clusters lead to similar or *more* diverse behaviors than 500 clusters, or would it collapse to similar performance as the no-clustering ablation? Similarly, it is not clear based on the ablation on the number of low-level steps that more than 100 steps (thus offloading more of the learning to behavior cloning rather than RL) would lead to worse behaviors. Given that the low-level policy is trained on a very large dataset, I imagine that behavior cloning over longer horizons, e.g., 1,000 steps, could still lead to very meaningful behaviors. Additionally, # of clusters and # of steps are highly dependent hyper-parameters given that they jointly balance how much of the behavior should be offloaded to the high-level RL policy vs. the low-level BC policy, but I didn't find any discussion or results that highlight this. Lastly, could the authors please clarify why the third ablation is conducted on Spider rather than the Log task like the two other ablations?
- The paper is lacking in terms of implementation details on the proposed method + the baselines. It would be helpful if the paper was more self-contained and described the overall architecture etc. in the appendix rather than simply referencing prior work. Additionally, in cases where baselines are adapted to new domains yet fail completely (e.g., VPT for the Kitchen environment), I would appreciate if the authors could share some thoughts on why this might be, even if not backed by data.

**Questions:**

I have a couple of additional questions that I would also appreciate if the authors could address:
- I understand the motivation behind the KL weight and that neither alpha=0 (no prior) nor too large of a weight are desirable. However, it appears that the authors choose to train the high-level policy from scratch and only leverage the goal prior to guide exploration. Given that the goal prior and high-level policy share the same action space, why do the authors decide against initializing the high-level policy as the goal prior and simply finetuning it using the proposed objective (reward + KL)?
- I am left wondering how many of the design choices in the proposed framework are uniquely beneficial for MineDojo due to its enormous state and action space. For example, are discrete goals really necessary for the simpler Kitchen environment compared to providing the raw physical state as a continuous goal? The MineDojo results are surely impressive, but it would be useful to contextualize the proposed method and design choices more for other domains besides MineDojo, with or without additional experiments to back any claims.

**Post-rebuttal:** I have revised my rating (6 -> 8) and confidence (3 -> 4) based on the authors' response to my comments, as well as those of my fellow reviewers.

---

> ### Author Response · Authors · 2023-11-20
> **Thanks for your review! Here, we respond to your comments and address the issues. We hope to hear back from you if you have further questions!**
>
> **Q1.** Ablations are fairly limited and leave several unanswered questions.
>
> **A1.** Thanks for your valuable suggestions to improve our ablation studies! We added ablation results in different number of clusters and low-level steps in Section 5.3. Results for both Log and Spider are presented in Appendix E and we will add results in all other tasks later.
> We summarize our conclusions below. More discussions can be found in Section 5.3 and Appendix E.
>
> For the choice of cluster numbers:
> - With N=10, the agent fails to improve the task success rates, due to the low capacity of behaviors in the small goal space.  But the task success rates are non-zero, due to the generalization ability of the low-level policy.
> - With an increasing number of goal clusters, the performance of PTGM is robust, while the sample efficiency and stability slightly decrease due to the larger high-level action space.
> - PTGM-no-cluster fails on all tasks, which is caused by ineffectiveness of RL acting in high-dimensional continuous action spaces. It can be viewed as the case where infinite number of goals are either in clusters or inexistent in the dataset.
>
> For the choice of the number of low-level steps:
> - With a small number of low-level steps (k=10), the behavior is offloaded more to the high-level policy, resulting in worse sample efficiency of RL.
> - With an increasing number of low-level steps, PTGM shows better sample efficiency and success rates. k=100 is a good choice for all the Minecraft tasks, where the high-level controller is able to switch about 10 goals per episode.
> - With k=500, PTGM converges to a lower performance compared with k=100 because the behavior is offloaded too much to the low-level policy. In this case, the high-level policy cannot switch many goals in an episode, making the task performance limited by the capability of the low-level controller conditioned on a single goal.
>
>
> **Q2.** Lacking implementation details on the method and the baselines. Explanation about the VPT baseline in Kitchen.
>
> **A2.** We enrich the implementation details of our method in Appendix A and the baselines in Appendix B.
> As explained in Section 5.2 (the "**VPT-finetune.**" paragraph), the VPT baseline in Kitchen is named BC-finetune, which pre-trains a behavior-cloning low-level policy and finetunes it with RL. Its failure is related to the lack of temporal abstraction for RL and the forgetting issue during RL finetuning. Pertsch et al. [1] report the similar results of this baseline in Kitchen.
>
> [1] Accelerating Reinforcement Learning with Learned Skill Priors, 2021
>
>
> **Q3.** Why don't we initialize the high-level policy with the goal prior in RL?
>
> **A3.** We conduct an ablation study that initializes the high-level policy with the goal prior model (PTGM-prior-init). As shown in Appendix E, Figure 11, PTGM-prior-init outperforms PTGM only on harvesting logs, but has worse performance in other Minecraft tasks.
> We find that since chopping trees is the most frequent behavior in the Minecraft Contractor dataset, the behavior-cloning models learned from such data can draw higher probabilities on harvesting logs than other behaviors. Thus, PTGM-prior-init is biased to the task of harvesting logs and fails to explore for other tasks in which the goal prior model draws low probabilities on the task-relevant goals. On the contrary, PTGM with random initialization of the high-level policy adopts a uniform goal prior for exploration, showing strong capabilities in solving out-of-distribution tasks.
> Since our problem formulation is to learn from **task-agnostic** data and finetune for arbitrary tasks, we believe that uniformly exploring in the goal space for RL is better.
>
> **Q4.** Are discrete goals really necessary for the simpler Kitchen environment?
>
> **A4.** We answer this question by implementing the action space of the high-level policy with the state space of Kitchen without clustering (PTGM-no-cluster). The results are presented in Appendix D, Figure 8.
> Though the dimension of the goal in Kitchen is relatively lower (21 dimensions), we find that PTGM-no-cluster fails in this task. We observe that, with the continuous action space, the high-level policy should learn to output valid goals that lay in the manifold of states from the dataset to make the low-level policy perform reasonable behaviors. On the contrary, for PTGM with discrete goal clusters, the output of the high-level policy is always a valid goal (in the clusters), making RL efficient. Though the KL reward encourages PTGM-no-cluster to output goals close to the goal prior, it cannot make the output accurate enough due to the Gaussian sampling of the action head.

---

> > ### Comment · Reviewer_NwB4 · 2023-11-21
> > **Thank you**
> >
> > Thank you for the detailed rebuttal, insights, and new experimental results. I recognize the amount of work that goes into running new ablations and baselines during such a short time window. I believe that the concerns in my original review have been addressed, and I have revised my rating (6 -> 8) and confidence (3 -> 4) accordingly.

---

> > > ### Author Response · Authors · 2023-11-22
> > > **Thanks!**
> > >
> > > Thank you very much for raising your score! We appreciate your constructive review.

---

### Meta-Review · Area_Chair_TNq3 · 2023-12-07

**Metareview:**

The authors propose a simple and elegant approach for the important problem of pretraining hierarchical policies from datasets of videos and actions without task rewards. The goal space is a clustering of all images in the dataset, a goal prior (e.g. Pertsch et al. 2011) is learned from the videos, and a low-level controlled is learned by goal-conditioned behavior cloning. Afterwards, a high-level policy is trained from environment interaction with task rewards, which is regularized towards the goal prior and whose goals are executed by the low-level controller. The empirical evaluation shows the promise of this approach. The authors are encouraged to release the source code and video of the agent with the camera-ready version. Future work should address the problem of learning goal states that handle partial observability and more sophisticated generalization.

**Justification For Why Not Higher Score:**

N/A

**Justification For Why Not Lower Score:**

Simplicity and potential scalability of the approach

---

### Decision · Program_Chairs · 2024-01-16

Accept (oral)